# Applying the Behaviour Change Wheel to UK Local Authority Policy Documents: A Content Analysis in the Context of Financial Behaviour

**DOI:** 10.3390/bs13120991

**Published:** 2023-11-30

**Authors:** Zach Mills, Katryn Wright, Danielle D’Lima

**Affiliations:** Centre for Behaviour Change, Department of Clinical, Educational and Health Psychology, University College London, London WC1E 7HB, UK

**Keywords:** behaviour change, financial behaviour, local government, financial well-being

## Abstract

Local authorities in the UK often try to improve their residents’ financial well-being by promoting changes in behaviour. The extent to which these behaviour change activities are based on relevant theory or evidence is unknown. This research aims to retrospectively analyse the content of local authorities’ policies to identify opportunities for improvement. The Action, Actor, Context, Target, Time (AACTT) framework was used to assess the specification of target behaviours. The Behaviour Change Wheel (BCW) process was used to assess intervention content. Within the policy documents, target behaviours were not consistently specified in terms of the AACTT criteria. Descriptions of interventions lacked detail with 28% unable to be categorised and there was a reliance on Education (46%) to change financial behaviour. The designing and reporting of interventions to change residents’ financial behaviour were not always aligned with behavioural science evidence and utilising systematic frameworks could help local authorities achieve policy objectives.

## 1. Introduction

The combined impacts of the COVID-19 pandemic and increases in the cost of living has meant that financial security is becoming increasingly difficult for households in the UK [1,2]. A state of financial security has been characterised as ‘financial well-being’. In the US, the Consumer Financial Protection Bureau (CFPB) define financial well-being as “a state of being wherein a person can fully meet current and ongoing financial obligations, can feel secure in their financial future, and is able to make choices that allow enjoyment of life” [3]. In 2020, the Money and Pensions Service (MaPS) launched the UK strategy for financial well-being defining it as “about feeling secure and in control. It is knowing that you can pay the bills today, can deal with the unexpected, and are on track for a healthy financial future. In short: confident and empowered” [4].

Both organisations highlight financial behaviour as the primary driver of financial well-being. The CFPB highlights four important behavioural areas: effective routine money management, financial research and knowledge seeking, financial planning and goal setting and following through on financial decisions [3]. The MaPS strategy highlights five key behavioural areas: getting a meaningful financial education, saving regularly, managing credit, accessing debt advice and making good decisions about future well-being [4]. Other behavioural areas such as meeting day-to-day spending commitments and financial product choices are also suggested to influence an individual’s financial well-being [5,6]. For the purposes of this research, ‘behavioural areas’ will refer to categories of discrete behaviours that are linked to a similar broad outcome.

It is important that, when defining the determinants of financial well-being, these behavioural areas are representative and relevant to households across the income spectrum. For example, in the UK, in 2017, GBP 10 billion worth of state benefits went unclaimed by those who were entitled to them [7]. The claiming of state benefits could have a considerable impact on the financial well-being of lower income households.

Within this research, three behavioural areas will be seen to represent key drivers of financial well-being. These are: individuals maximising their income by claiming the financial support available, individuals paying regular bills on time and individuals saving regularly to protect against income shocks. These areas are consistently highlighted as important factors in the financial well-being literature and are likely to be relevant to households across the income spectrum [3,4,5,6]. These will be referred to as ‘behavioural areas of interest’.

In the UK, local authorities are in a unique position to influence the financial well-being, and associated behavioural areas, of their residents. They have daily interactions with residents’ lives whilst also having a direct financial relationship by collecting council tax payments, administering some benefits and, for some, acting as the landlord [8]. In recognition of this, local authorities often develop policies and strategies to outline the actions they are going to take to support their residents towards better financial outcomes whilst ensuring continued collection of revenue [8]. These policies often outline funding and service provision commitments as well as describing activities to encourage particular behaviours within residents (also known as “interventions”) (e.g., council tenants paying their rent on time, residents paying council tax via direct debit, or residents applying for local and national financial support schemes). For example, in 2021, Maidstone Borough Council launched their Financial Inclusion Strategy 2021–2026. This outlined some broad aims to improve the living conditions and financial circumstances of their residents as well as aims to change resident behaviour to help drive these improved circumstances (e.g., “increase awareness and take up of mainstream financial services amongst low-income households”) [9].

The policies and strategies that local authorities publish represent their, publicly available, documentation of the planned activities to change residents’ financial behaviour. Policies are highlighted as a means of generating and guiding the implementation of behaviour change interventions [10,11]. Therefore, the content and specification of policies are suggested to be extremely important in the later implementation of interventions and the associated outcomes [12,13].

It is suggested that any intervention aimed at changing behaviour should be informed by relevant theoretical and empirical evidence [14,15,16]. Theory can be used to inform interventions by highlighting constructs that are hypothesised to have causal relationships with the behaviour and therefore represent relevant targets for intervention [17]. The utilisation of empirical evidence allows for a more informed assessment of whether an intervention is likely to be effective across different contexts, populations and behaviours [16].

A crucial first step in the design of interventions is the specification of a behavioural target [18]. A clearly specified behaviour can help to highlight the various people at different levels that need to do something differently, identify any barriers and enablers of the behaviour and design interventions to overcome them whilst providing metrics for evaluation [19,20,21]. Although research into the content of local authority policy documents is limited, it has been suggested that the design and specification of some policies do not allow for clear monitoring or evaluation of outcomes [22].

Theories, models and frameworks can aid both the specification of target behaviours and design of interventions [20,23,24,25]. The use of these tools can help intervention development via two main ways: more effective design and clearer reporting [14,15,26].

The use of systematic theories and frameworks aids more effective design of interventions by enabling policymakers and practitioners to build on an existing evidence base [26]. A systematic approach also creates a common language for characterizing intervention activities, which helps to prevent false innovation where new labels are used for existing theoretical constructs [26]. Theories and frameworks help to utilise important intervention options that could have been otherwise missed [14,15].

Accurate reporting of behaviour change interventions is important for understanding their effectiveness [27]. The inconsistency of reporting behaviour change interventions has been recognised [28], leading to calls for more detailed descriptions of intervention content, delivery and implementation [29]. In health behaviour, The Consolidated Standards for Reporting Trials (CONSORT) Statement was developed to improve the design and reporting of interventions [30].

Policy documents are often the initial, and sometimes only, record of behaviour change interventions and can be used to guide their later implementation [10]. Therefore, it could be argued that the design and reporting within policy documents should be approached using the same systematic methods. This could increase the effective implementation of interventions, allow for clearer policy evaluation and contribute to the evidence base of what works, when and for whom.

To current knowledge, there has been no prior research analysing the content of local authority policy documents using frameworks informed by behavioural science. This type of analysis is important for two main reasons. Firstly, it would be an important first step in identifying any gaps between the evidence base within behaviour change research and the current practices of local authorities in terms of their designing of behaviour change interventions and reporting within policy documents. It also empowers policymakers through an enhanced understanding of the theoretical and empirical foundations of the policies and behaviour change activities they implement and provides a roadmap to enhancing effectiveness.

Secondly, this type of analysis helps to advance understanding of how behavioural science frameworks can be used to analyse a wide range of policy documents, specifically in the context of financial behaviour. Behaviour change theories and frameworks have started to become more widely used to design interventions and retrospectively analyse policies in health settings [11,21,31]. However, their use has been less common in the context of financial behavior and financial well-being. This research aims to understand the transferability of these theories and frameworks to novel contexts.

The Behaviour Change Wheel (BCW) is a framework that could guide this type of analysis. The BCW is a theory-based framework used to characterise and design behaviour change interventions [14,15]. At its core, the BCW uses the COM-B model (capability (C), opportunity (O), motivation (M) and behaviour (B)) to first understand the sources of the behaviours that could be useful targets of intervention. The COM-B model suggests that an individual’s physical and psychological capability (e.g., physical skill, strength, psychological skills or knowledge), social and physical opportunity (e.g., interpersonal influences, cultural norms, time or resources) and reflective and automatic motivation (e.g., plans, beliefs about what is good and bad, emotional reactions, desires and impulses) all interact to generate behaviour [14,15]. Therefore, any interventions should target the sources of behaviour that are most likely to be effective in the given context [14,15].

Within the BCW process, one of the first, and often overlooked, steps in designing a behaviour change intervention is to select and specify a target behaviour. Specialised frameworks, such as the Action, Actor, Context, Target, Time (AACTT) framework [20], have been developed to guide the specification of target behaviours.

Once a target behaviour has been specified, the BCW recommends performing a behavioural diagnosis to understand what factors could inhibit (known as barriers) or facilitate (known as enablers) the performance of the behaviour. The barriers and enablers can be categorised under the three main sources of behaviour within the COM-B model to help select the most suitable interventions options [14,15].

The BCW categorises potential intervention activities into nine broad intervention types (ITs). These are Education, Persuasion, Incentivisation, Coercion, Training, Restriction, Environmental Restructuring, Modelling and Enablement. It also presents seven policy categories that can support or enable these activities [14,15]. Intervention content can be specified further through the use of a Behaviour Change Technique Taxonomy (BCTTv1) which contains 93 ‘Behaviour Change Techniques’ (BCTs) [32]. For example, an education-based intervention may include BCTs like providing feedback on the social and environmental consequences of a behaviour and instructions on how to perform the behaviour.

### 1.1. Literature Review

Previous research has used the AACTT framework to evaluate the descriptions of behaviours within health policy documents [21]. It was found that the Actor, Target, Time and Context were often poorly specified in the documents analysed [21]. The framework has also been used to understand the level of specification of behaviours within pharmacy practice standards [31]. It was found that the Actor was never specified within these documents and the Action, Context, Target, and Time were often poorly and inconsistently specified. This research from other policy settings suggests that the framework can be adapted and applied to retrospectively analyse the specification of behaviourally relevant content within documents [21,31].

The BCW process has previously been used to ‘retrofit’ planned health interventions [33] and has demonstrated its utility in characterising existing interventions. It was also used to analyse health policy documents [11] to understand the types of interventions, their content and the mechanism through which they could have an effect. The research found that descriptions of recommendations were sometimes too vague to confidently identify the type of intervention or its content. When it could be identified, the recommendations were mainly trying to influence behaviour through Psychological Capability, Physical Opportunity and Social Opportunity [11]. The most commonly identified ITs were Environmental Restructuring, Education and Enablement.

A literature review did not identify any examples where either the AACTT framework or BCW process had been used to analyse local authority policy documents. There was also no prior research identified where any types of policy documents were analysed within the sequential intervention design steps of the BCW process.

### 1.2. The Current Research

This research will use the AACTT framework [20] to assess the extent to which the target behaviours of interventions, to influence residents’ financial behaviour, are specified within local authority policy documents. The BCW [14,15] will be used to guide the analysis of any reported barriers or enablers of relevant behaviour, the types of interventions being described, the content of those interventions and the mechanisms through which they could be having an effect. The selection of interventions will be assessed on their suitability to influence the recognised barriers and enablers.

#### Research Questions


Research Question 1 (RQ1): To what extent are the target behaviours of interventions, reported in the documents, specified in terms of Action, Actor, Context, Target and Time (AACTT framework)?


The clear specification of a target behaviour is a critical first step in designing an intervention. It allows for a better understanding of the factors that are driving or facilitating that behaviour in a given context and what might need to be changed. This aids a more effective selection of the most appropriate intervention for changing that behaviour [18,19,20,21]. Local authority policy documents are likely to be read and interpreted by a wide range of people; therefore, the clarity of target behaviours is particularly important to ensure intervention activities are designed and implemented as consistently and effectively as possible. This research will use the AACTT framework [20] to assess the extent to which the target behaviours of interventions, to influence residents’ financial behaviour, are specified within local authority policy documents.
Research Question 2 (RQ2): Which sources of behaviour (capability, opportunity and/or motivation) are cited as barriers or enablers of relevant financial behaviour?

A behavioural diagnosis is an important step when designing an intervention, as it helps to identify factors that may be inhibiting (barriers) or facilitating (enablers) a behaviour. Reporting the perceived behavioural barriers and enablers in a policy document helps to create shared understanding and allows for a more effective selection of interventions in a given context. This research will use the COM-B model to identify the different sources of behaviour that are explicitly mentioned as barriers or enablers of relevant behaviours.
Research Question 3 (RQ3): Which types of interventions, recognised by the BCW process, feature within the documents?

A clear specification of the target behaviour and explicit behavioural diagnosis can aid more effective selection of interventions [19,20,21]. This research will aim to, where possible, categorise the interventions described in the policy documents in line with the nine ITs included in the BCW. This will provide insight on how well the selection of interventions align with the current theory and evidence within behavioural science.
Research Question 4 (RQ4): Which behavioural mechanisms could the selected interventions effect according to the BCW matrix?

The BCW includes a matrix which links the nine ITs to the COM-B components. This helps to explain the mechanisms through which each intervention could have an effect. Using this matrix, the current research will look to retrospectively identify the behavioural mechanisms that the interventions, described in the policy documents, could be targeting. This will provide insight on how well the selection of interventions align with theory and evidence within behavioural science and how effective they could be at influencing any explicitly mentioned barriers or enablers of that behaviour.

## 2. Methods

### 2.1. Design

This study used a directed content analysis design [34]. A content analysis was chosen because it is a well-established and widely used method of systematically analysing textual data and is therefore suitable for the examination of policy documents. A directed content analysis reflects a deductive approach which allowed for the use of existing theoretical frameworks to develop pre-determined categories for data coding. The deductive approach ensured that the analysis was structured and guided by established theoretical frameworks. This methodology was important for evaluating the content of policy documents and alignment with behavioural science frameworks. Figure 1 gives an overview of the key stages involved in conducting this research.

### 2.2. Document Search

A systematic search process was developed to efficiently identify local authority policy documents containing information regarding their activities related to the three behavioural areas of interest (i.e., residents claiming financial support, paying bills, and saving). The official websites of all English district, borough and city councils as well as unitary tier local authorities were searched. The steps involved in the search process are outlined in Appendix A.

Key search terms were used within the search bar of each local authority website to identify potentially relevant documents. The search terms were developed through a piloting exercise where a variety of local authority policy documents were examined, and a judgement was made on their relevance to the research questions. The most common terms within the names of relevant documents were used as search terms. These were “Financial Inclusion”, “Poverty”, “Anti-Poverty”, “Income” and “Rent”. Documents that were identified through this search and met the following initial criteria were deemed to warrant a further search for relevant content: (1) the document must be published officially by the local authority; (2) it must be outlining a policy or strategy that the council has committed to (e.g., not a set of recommendations from an external organisation or commissioned research); (3) it must represent a live policy or strategy that is currently in place (i.e., it covered the year of the search, 2022). If no date was present on the document but it was the most recent version on the website, then it was included.

The documents that met this criterion were searched for potentially relevant content using a second set of key terms: “rent”, “council tax”, “income”, “benefits”, “take up”, “income maximisation”, “bills”, “saving” and “save”. These search terms were again developed through a piloting exercise where common terms within relevant sections of the document were collected and assessed. The terms were used as a method of navigating to the relevant sections of the document and did not impact data extraction. If the document contained any information relevant to any of the three behavioural areas of interest, then it was included in the final sample for data extraction. See Table 1 for a more detailed description of the three behavioural areas of interest.

### 2.3. Data Extraction

A data extraction form was developed to capture the following document characteristics: local authority, name of the document, and the years the policies were in place. The form also captured whether the data had been extracted because it met the criteria to be analysed under RQ1, RQ2 or RQ3/RQ4. Table 2 shows the criteria that data fragments had to meet to be extracted for analysis. The data fragments were only extracted if residents were judged to be the target population rather than council employees or any other group. The data fragment had to meet all the criteria, for at least one research question, to be extracted.

### 2.4. Analysis

#### 2.4.1. Coding Form

A coding form was developed (see Appendix A) and fragments of data were transferred from the extraction form to the coding form to perform the analysis. The coding form had a row for each data fragment and a column for each coding criterion. Standardised drop-down options were used to assign codes which helped with frequency calculations. Codebooks were developed to guide the coding of each criterion to data fragments. The development of the codebooks is explained in more detail below.
RQ1: To what extent are the target behaviours of interventions reported in the documents specified in terms of Action, Actor, Context, Target and Time (AACTT framework)?

The AACTT framework [20] was used to assess the specification of target behaviours within the extracted data. The definitions for each domain (e.g., ‘Action’) were adapted from previous research [31]. Table 3 shows the codebook used to assign codes to the descriptions of target behaviours. If the data fragment did not contain a description of a target behaviour (i.e., because it had met the extraction criteria for RQ2 or RQ3/RQ4 but not RQ1) the fragment was coded as “No” within a column labelled “Has a target behaviour been specified?” and the AACTT criteria were not applied.

The criteria for the ‘Action’ domain within the AACTT framework required an assessment of whether the Action could be considered discrete and observable. Discrete Actions are said to be “a single self-contained action without consideration to whether there are ancillary actions that may precede or proceed on from that single self-contained action. A non-discrete action may be the result of a series of discrete actions though were considered non-discrete if it is stated without further description of the process. The assessment of whether the action is discrete includes an assessment of whether the reader is provided with the full details of the action or expected to bring pre-existing knowledge to interpret the action in a consistent manner” [31].

Whereas, an observable Action is “specified as an externally visible manifestation and/or the direct outcome of the action would be a physical object. Non-observable actions are those where there is no external physical sign that an action is performed” [31]

To support the consistent application of the codebook, and following the methodology used in previous research, ambiguous verbs were assessed, alongside their dictionary definitions, to indicate whether their use indicated discrete and observable actions [31]. A record of the assessment made for each Action word can be found in Appendix A. For example, “take-up” was used to describe a target behaviour of residents within the documents (e.g., “We will encourage residents to take up the benefits they are entitled to”), but it was considered to be neither a discrete nor observable Action. Therefore, it was coded as “No” (i.e., ‘Action’ criteria not met) whenever it appeared.
RQ2: Which sources of behaviour (capability, opportunity and/or motivation) are cited as barriers or enablers of relevant financial behaviour?

A codebook was adapted from previous research to identify when barriers or enablers of relevant behaviours could be categorised under any of the COM-B components [11]. Table 4 shows the definition used, a general example and the coding criteria for each COM-B component. The barriers and enablers had to be explicitly recognised within the document and could not be inferred from the content of an intervention. For example, “People may not have the right knowledge or resources to make a claim for Universal Credit” explicitly recognises a lack of knowledge as a barrier to claiming Universal Credit.

If no barriers or enablers had been explicitly described, then it was coded under “No” in the column “Have any barriers/enablers been specified?”. If a barrier or enabler had been recognised but it was not described in enough detail to code to any COM-B component, then it was coded under “Too vague to be coded”.
RQ3: Which types of interventions, recognised by the BCW process, feature within the documents?

A codebook was developed to code occurrences of the nine ITs outlined in the BCW process. A data fragment could be coded to multiple ITs if it was judged that more than one was present. Table 5 shows the definitions of each IT, alongside a general example and an example relevant to the research context.

If a data fragment did not contain any description of an intervention, then it was coded as “No” in the column “Has an intervention been specified?”. If an intervention was present but the description was not explicit enough to code under any of the ITs, it was coded as “Too vague to be coded”.

The descriptions of intervention content was also coded to identify any possible BCTs using the BCTTv1 [32]. The BCTTv1 contains 93 BCTs; therefore, it was not considered feasible to create relevant examples for each. Within the taxonomy, the BCTs are organised into clusters (e.g., feedback and monitoring, shaping knowledge) which were used to narrow down the search for potentially relevant BCTs. For example, if an intervention was coded to Education, then the feedback and monitoring and shaping knowledge BCT clusters were checked first to see if any were present in the intervention.

If a description of the intervention content was not detailed enough to code for any BCTs then this was coded as “No” in the column “Is the intervention able to be categorised using BCTTv1?”. All data fragments that were coded as “No” in the column “Has an intervention been specified?” were not assessed for BCTs. The full coding instructions for BCTs can be seen in Appendix A.
RQ4: Which behavioural mechanisms could the selected interventions effect according to the BCW matrix?

Within the BCW process, there is a matrix used to link sources of behaviour (recognised by the COM-B model) to congruent ITs (see Table 6). This matrix can be used to identify suitable ITs following an analysis of the possible barriers or enablers of the chosen target behaviour. This matrix was used in reverse within this research. The ITs that had been identified were mapped back to all the possible behavioural mechanisms through which they could have an effect [14,15].

For example, an intervention coded to Education was mapped back to Psychological Capability and Reflective Motivation, as according to the BCW matrix, the intervention could be affecting behaviour through one or both of these mechanisms.

#### 2.4.2. Pilot Coding and Reliability Checks

To aid the accuracy and consistency of the coding, a pilot coding exercise was conducted before finalising the codebooks. The first coder (ZM) coded approximately 16% (*n* = 39) of the total sample of data fragments. The pilot coding was used to ensure a full understanding of all the criteria within the frameworks. A second coder (KW) then reviewed the same sample of data fragments.

The first and second coder discussed any differences in their coding and attempted to reach a consensus. For data where the coding was still ambiguous, a third coder (DD) was consulted before iterations to the code book were made. For example, for interventions where a new advice service had been provided for residents, there was some ambiguity as to whether this should be coded to the Education, Environmental Restructuring and/or Enablement ITs. Following consultation with the third coder, it was decided that these interventions should be coded to Education and Environmental Restructuring as, from the descriptions, there was no clear link to Enablement. Descriptions of all the data fragments that were referred to the third coder can be seen in Appendix A.

The first and second coders (ZM and KW) were selected due to their postgraduate level training on the frameworks used in this research and application using qualitative methods. The third coder (DD) is a senior behavioural scientist with experience of applying the frameworks in this context and training others to do so.

## 3. Results

### 3.1. Document Search

A total of 316 local authority websites were searched using the search process outlined. In total, 22 policy documents from 21 different local authorities met all the inclusion criteria and were included in the final sample. The types of documents that were included can be categorised into three broad areas: (1) promoting financial well-being (e.g., financial inclusion strategies, financial well-being strategies), (2) tackling poverty (e.g., anti-poverty strategies, poverty reduction strategies) and (3) council income management (e.g., rent arrears policy, debt management strategies). A list of the documents included in the final sample can be seen in Appendix A.

### 3.2. Data Extraction

The number of data fragments that were extracted per document ranged from 3 to 29. A total of 238 data fragments were extracted from the 22 documents. For example, “Encourage low-income residents who are claiming either Working Tax Credit, Child Tax Credit or Universal Credit to save by promoting the Help to Save savings account” (Blackpool, Financial Inclusion Strategy) was extracted, as it includes a description of a target behaviour (saving) and a description of an intervention (promoting the Help to Save savings account) related to one of the behavioural areas of interest (saving). Whereas, “Emergency support can be provided to help with daily living costs, housing-related costs and household items” (East Riding of Yorkshire, Financial Inclusion Strategy) does not describe a target behaviour, barriers or enablers, or intervention related to one of the behavioural areas of interest, and therefore was not extracted. See Appendix A for further examples of data fragments that were and were not extracted from the documents.

The number of data fragments related to each behavioural area of interest can be seen in Table 7.

### 3.3. Analysis

#### Pilot Coding and Reliability Checks

A reliability score was calculated for AACTT (60%), ITs (67%), BCTTv1 (80%) and COM-B (80%) individually. The combined reliability score was 73%.
RQ1: To what extent are the target behaviours of interventions, reported in the documents, specified in terms of Action, Actor, Context, Target and Time (AACTT framework)?

A total of 131 of the data fragments included descriptions of target behaviours. Table 8 shows the number and percentage of the descriptions of target behaviours that met each coding criteria within the AACTT framework.

***Action.*** None of the descriptions of target behaviours were judged to meet the Action criteria. This was because the descriptions were often not specific or detailed enough to identify the discrete Action. For example, “The housing officer will meet the prospective tenant and advise them to apply for Universal Credit or Housing Benefit” (Dartford, Rent Arrears Policy). The target behaviour of this intervention is judged to be the tenant applying for Universal Credit or Housing Benefit. However, without more information given on how the tenant should do this, it does not meet the Action criteria.

***Actor.*** Within 99 (76%) of the descriptions of target behaviours, an Actor was specified. The Actor was often described as “the resident” or “the tenant” which was judged to be sufficient given the context of the documents. The descriptions that did not specify the Actor required an assumption or pre-existing knowledge from the reader (e.g., “encourage payment by Direct Debit”). Other descriptions made some reference to non-specific groups (e.g., “individuals”) which would not be useful in determining who was supposed to be performing the behaviour and did not meet the criteria.

***Context.*** The Context in which the behaviour should be performed was described sufficiently to meet the criteria in four examples (3%). All four of these descriptions involved the council specifying the methods through which a resident could pay their rent (e.g., “There are a range of methods through which rent payments can be made including direct debit, online, standing order, over the telephone and all pay cards” Hinckley and Bosworth, Rent Arrears and Recovery Policy).

A further eleven examples (8%) included some attempt at describing the Context, but more information was needed to establish the exact setting or location the residents would be performing the behaviour (e.g., “Work with GP surgeries and walk-in-centre services to support residents in completing application forms and to promote signposting to current service providers” Blackpool, Financial Inclusion Strategy).

***Target.*** The Target was not deemed necessary in the context of this research and the documents being analysed. The behaviours of residents were unlikely to be on behalf of anyone else other than themselves. A possible exception would have been if the resident was expected to behave on behalf of another household member, but examples of this were not identified within the documents.

***Time.*** The Time at which the resident would perform the behaviour was described in sufficient detail in four examples (3%). This was often when the document was describing when residents were expected to pay their rent (e.g., “Tenants who pay their rent weekly, need to pay on the Monday each week and tenants who pay it on a fortnightly or monthly basis need to pay in advance” East Devon, Income Management Strategy).

In a further 16 examples (12%) there was some reference to the Time, but more detail was needed. This was often when the description had indicated that residents should pay their rent “on time” or seek advice “at an early stage”. This gives some broad indication as to when a resident should perform the given behaviour but there could be multiple interpretations.

Table 9 provides illustrative examples of possible amendments that could be made to the descriptions of target behaviours to meet each specification criteria within the framework.
RQ2: Which sources of behaviour (capability, opportunity and/or motivation) are cited as barriers or enablers of relevant financial behaviour?

A total of 8 out of the 22 documents, included in the final sample, included descriptions of barriers or enablers of behaviours within the areas of interest. Therefore, only 26 data fragments explicitly mentioned barriers or enablers. Of these, 22 were described in enough detail to confidently code under any of the COM-B components. When a barrier or enabler had been identified but was not able to be coded under a COM-B component, this was often because there was not enough detail about what the specific barrier or enabler actually was. For example, “Lone parents face issues around the use of banking services, low levels of savings and debt” (East Riding of Yorkshire, Financial Inclusion Strategy) recognises that lone parents face issues (and therefore barriers exist), but does not describe what these are or why they have an effect on their use of services and saving.

Table 10 shows the number of data fragments that could be coded to each COM-B component alongside an example of each. Some data fragments could be categorised under more than one COM-B component. For example, “People may not have the right knowledge or resources to identify the best product for their financial circumstances” (East Riding of Yorkshire, Financial Inclusion Strategy) was coded under Psychological Capability and Physical Opportunity.
RQ3: Which types of interventions, recognised by the BCW process, feature within the documents?

A total of 185 of the extracted fragments of data contained a description of an intervention.

***ITs.*** Table 11 contains an overview of the number and percentage of data fragments that could be coded to each of the nine ITs, alongside an example from the research. Data fragments would often be coded to more than one IT if they were judged to be present. For example, “The Rental section will work in partnership with agencies to facilitate the provision of independent advice to tenants about money management” (East Devon, Income Management Strategy) was coded to both Education and Environmental Restructuring.

***BCTs.*** There were 81 data fragments (44% of the data fragments that contained a description of an intervention) where BCTs could be identified. Where BCTs could not be identified (n = 104, 56%) within the data fragments this was due to a lack of granular detail regarding the content of the intervention. All interventions where the IT was not able to be identified were also not coded to any BCTs.

There were 14 different BCTs identified across all the data fragments. Table 12 provides an overview of the number and percentage of data fragments where each BCT was present, alongside an example from the research.
RQ4: Which behavioural mechanisms could the selected interventions effect according to the BCW matrix?

All of the sources of behaviour could have been influenced by the described interventions. Almost two-thirds (61%) of the described interventions could have influenced the Psychological Capability of residents. Additionally, almost half (47%) could have been influencing the Reflective Motivation of residents. Table 13 gives an overview of the number and percentage of interventions that could be influencing behaviour through the different COM-B components.

## 4. Discussion

This research used the AACTT framework [20] and BCW process [14,15] to perform a retrospective content analysis of local authorities’ documented activities to influence behaviour within three key behavioural areas of financial well-being (claiming financial support, paying bills, and saving). The research aimed to highlight any gaps between the evidence within behavioural science and current practices by local authorities, in terms of their designing of behaviour change interventions and reporting within policy documents.

The findings demonstrate that the target behaviours of interventions could be specified further within the documents to be closer aligned with reporting frameworks within behavioural science. There was no description of a target behaviour where the Action, Actor, Context and Time could all be confidently identified. Additionally, only a small number of the documents made any explicit references to potential behavioural barriers or enablers. The descriptions of interventions to influence behaviour often did not include sufficient detail to confidently identify their type and content. When the type of intervention could be identified, there appeared to be reliance on Education to change residents’ financial behaviour within the areas of interest. Finally, the mechanisms of action, through which the selected interventions could influence behaviour, were not necessarily the most suitable to address the recognised barriers and enablers of relevant behaviours.
*RQ1: To what extent are the target behaviours of interventions, reported in the documents, specified in terms of Action, Actor, Context, Target and Time (AACTT framework)?*

The lack of specification of target behaviours in terms of Action, Actor, Context and Time (Target was not considered necessary in this context) is consistent with the limited previous research that has explored the specification of target behaviours in policy documents [21] and pharmacy practice standards [31].

The findings show that the Actor (i.e., the person who should perform the behaviour) was most often identified and met the criteria within the AACTT framework in 76% of the descriptions. Interestingly, this differs from previous research where the Actor was never specified within pharmacy practice standards [31]. Within practice standards, it is likely that the intended reader is also the intended Actor of the behaviours being described. Therefore, it may have seemed less relevant to include specification of the Actor. Within the local authority policy documents, there was frequent reference to “residents” and “tenants” which was deemed sufficient to meet the criteria. However, in a practical guidance document, a more specific description of the Actor may be useful.
*RQ2: Which sources of behaviour (capability, opportunity and/or motivation) are cited as barriers or enablers of relevant financial behaviour?*

Most reported barriers and enablers of behaviours within the areas of interest were related to Physical Opportunity or Psychological Capability. This appears to suggest that these local authorities recognise that time and other physical resources (Physical Opportunity) plays an important role in residents performing behaviours relating to claiming financial support, paying bills and saving. Additionally, the knowledge and mental capacity of residents (Psychological Capability) are seen to be important.

To current knowledge, there has been no previous research that utilised the COM-B model to identify any barriers or enablers of targeted behaviours within policy documents. This was a novel feature of this research.
*RQ3: Which types of interventions, recognised by the BCW process, feature within the documents?**ITs*

The descriptions of interventions often lacked detail which meant their type and content was not always able to be identified. When the type of intervention could be identified, there was a reliance on Education as a means of changing behaviour. Many of the interventions also had elements of Environmental Restructuring and Enablement. The other six ITs were identified rarely or not at all.

These findings overlap with previous research with some interesting differences. Within the health policy documents analysed in previous research, the descriptions of interventions were also not always sufficiently described to identify if any of the nine ITs were present [11]. Within the health policy documents, the most commonly identified IT was Environmental Restructuring, followed by Education and Enablement [11]. All of the nine ITs were identified at least once within the documents analysed.
*BCTs*

The lack of granular detail in intervention content meant that BCTs were only able to be identified in under half (44%) of the data fragments that included a description of an intervention. When the content of interventions could be identified, there was a lack of variety. Only 14 (out of a possible 93) different BCTs could be identified, with practical social support and adding objects to the environment being the most prominent.

These findings differ from previous research where a wider variety (35 out of a possible 93) of BCTs were identified within health policy documents [11]. The nature of the BCTs also differed, the health policy documents reported interventions that put more emphasis on giving information about health consequences and instructions on how to perform the behaviour. This is to be expected as the interventions are targeting very different behaviours. There was some overlap, however, with both sets of policy documents including social support and restructuring of the physical environment [11].

The better specification and more diverse selection of interventions and intervention content within health policy documents could be due to more advanced knowledge and utilisation of behavioural science frameworks. There is considerably more literature regarding the explicit use of systematic behavioural science frameworks within health settings [16,19,33] compared to local authority settings. This could be reflected in the content and reporting of behaviour change interventions and policies.
*RQ4: Which behavioural mechanisms could the selected interventions effect according to the BCW matrix?*

All of the sources of behaviour could have been influenced by the described interventions, with Psychological Capability and Reflective Motivation being the most commonly identified. These findings differ from previous research where Physical Opportunity and Social Opportunity were the most commonly identified mechanisms of action through which the interventions could be having an effect [11].

The mechanisms through which the selected interventions could have an effect do not align perfectly with the barriers or enablers that were recognised as influences on relevant behaviour. Physical Opportunity was recognised as the most likely barrier or enabler; however, only 38% of the interventions could influence Physical Opportunity. Additionally, 47% of the selected interventions could influence residents’ Reflective Motivation; however, this was never explicitly highlighted as a likely barrier or enabler in any of the documents. On the other hand, Psychological Capability was the second most recognised barrier or enabler and could be influenced by 61% of the interventions selected.

To current knowledge, the assessment of the suitability of selected interventions to influence the recognised barriers or enablers of behaviour with policy documents was also a novel feature of this research.

### 4.1. Implications

This research has some practical implications for the design, reporting and implementation of local authorities’ policy documents in the context of financial behaviour. Firstly, the research highlights that the target behaviours, barriers and enablers, and interventions are not always reported in sufficient detail to confidently categorise them using recognised behavioural science frameworks. Regardless of the intended audience or purpose, any documentation of behaviour change interventions benefits from clear reporting [20,27,29]. As discussed, a clearly specified target behaviour helps to highlight the various people at each level that need to do something differently, is a crucial first step in identifying the barriers and enablers and provides an indicator of what to measure to evaluate an interventions effectiveness [19,20,21]. Clear reporting of intervention content within policy documents could also help local authority practitioners implement consistent interventions, allow for clearer policy evaluation and contribute to the evidence base of what works, when and for whom [12,13,20,25].

The use of systematic frameworks could help to create common terminology and reporting standards for local authorities’ policy documents and the interventions to change residents’ financial behaviour [26]. This could help to improve the dissemination and interpretation of interventions and their outcomes within and across different local authorities further adding to the evidence base of works, when and for whom.

Secondly, this research has highlighted that the selection of interventions reported within policy documents does not represent the full range of intervention options recognised as being effective in changing behaviour. The use of systematic frameworks to design behaviour change interventions can help policymakers and practitioners utilise important intervention options that could have been otherwise missed and select the most appropriate intervention options to influence the key barriers and enablers of behaviour [14,15]. Additionally, the behavioural mechanisms through which the selected interventions could have an effect are not necessarily the most suitable to influence the reported barriers or enablers. A systematic framework could help local authorities create a clear and evidence-based, audit trail within policy documents from the identification of barriers and enablers to the selection of potentially effective intervention options [14,15].

In addition to the practical implications, this research also has theoretical implications for the field of behavioural science. Firstly, this research has highlighted that the BCW is a useful tool for systematically analysing descriptions of behaviour change interventions outside of the health context. It can help to create a common understanding and shared language to categorise interventions reported within policy documents. This research has also identified where the BCW approach is not always sufficient to accurately characterise the intervention activities reported in local authority policy documents. It could be beneficial to further differentiate some of the broader ITs (e.g., Environmental Restructuring) or BCTs (e.g., practical social support) to allow for more precise specification of activities. For example, interventions that included the provision or adaptation of a service to help residents would be categorised under Environmental Restructuring. However, this does not paint a detailed picture as to the different activities that could be occurring on the ground as a result of these changes in the environment. Further specification would allow for more accurate reporting of intervention activities and strengthen understanding of what works.

Secondly, this type of research could be used to contribute to the science of behaviour change and inform future refinements of behavioural science theories, models and frameworks. For example, a similar content analysis of policy documents could be conducted alongside an evaluation of the intervention strategies. This would help to understand if the interventions have the predicted effects on behaviour and the extent to which these effects occur via the hypothesized mechanisms of change. These findings could be used to inform updates to the BCW matrices, for example, which represent the hypothesised links between intervention types and mechanisms of change.

### 4.2. Strengths and Limitations

To the researchers’ knowledge, this is the first study that has applied any behavioural science frameworks to analyse the content of local authority policy documents aimed at changing resident financial behaviour. This could have caused issues regarding the interpretation of novel data and meant the coding criteria were open to a high degree of subjectivity. However, the researchers ensured rigour in the process by firstly adapting an initial version of the codebook from similar research where the frameworks had been used to analyse documents in different policy settings. The codebook was adapted for this research to include clear definitions, generic and context-specific examples, and instructions to aid the consistency of coding. A pilot coding and reliability coding exercise was also undertaken which helped to sense check the researchers’ interpretation of the data and coding criteria. There was space within the methodology to iterate the codebook following this process.

A limitation of this research is that the documents used did not perfectly align with the behavioural areas of interest. The documents were not created to specifically or exclusively promote behaviours related to residents claiming financial support, paying their bills and savings. This is reflected by the large amount of data related to paying bills, rather than claiming financial support or saving. Local authorities act as landlords for council tenants and collect council tax from residents and are, therefore, more likely to have policies and strategies within this area. These behavioural areas of interest were chosen as points of investigation due to their important role in determining financial well-being. However, it is possible that not all of the activities that councils were undertaking to influence behaviour within these areas would be reported in the documents analysed.

A second limitation was the methodology used to identify the mechanism of actions within the described interventions. The BCW matrix identifies all of the possible mechanisms through which the interventions could be having an effect [14,15]. The linking of COM-B components to the ITs was proposed by experts in the field of behaviour change [14,15], but it does not necessarily reflect the actual mechanisms through which interventions are having their effect (or lack thereof) in a given context.

### 4.3. Future Research

Future research could look to replicate this type of content analysis on policy documents that focus on other key behavioural areas of financial well-being, as highlighted by CFPD and MaPS. For example, the MaPS suggested ‘accessing debt advice’ as an important behaviour [4]. Many local authorities develop debt management policies to outline how they are going to collect a debt owed to them (e.g., rent or council tax arrears). These often include descriptions of interventions designed to encourage specific behaviours such as accessing debt advice and making re-payments. The documents could benefit from an analysis to see how well the interventions are reported and where the gaps are between the evidence within behavioural science and current practices.

It would also be useful to conduct research with policymakers in local government to explore the current barriers and enablers of using behavioural science frameworks in the development of policies and strategies. This could be used to inform any necessary adaptions to the frameworks to improve their usability for policymakers in different settings.

## 5. Conclusions

This research has shown that descriptions of target behaviours could be specified further within local authority policy documents to align with reporting frameworks within behavioural science. Local authorities rarely describe potential barriers or enablers of these financial behaviours within their policy documents. This makes it harder to build a common understanding of the key factors that need addressing to change behaviour and could result in the design and implementation of interventions targeting inappropriate and ineffective sources of behaviour. The descriptions of interventions often lack detail which can make it difficult to identify the type of intervention being utilised and the activities involved. This could limit local authority practitioners’ ability to implement consistent interventions, inhibits policy evaluation and makes it harder to build an evidence base of what works, when and for whom. When intervention content can be identified, councils appear to often use Education as a means to change residents’ behaviour regarding claiming financial support, paying their bills or saving. The interventions selected by councils were not necessarily the most suitable to influence the barriers or enablers reported within the documents.

The research has highlighted the current gaps between the evidence within behavioural science and the current practices of local authorities in their design of behaviour change interventions and reporting within policy documents. The use of systematic frameworks to guide the design and reporting of behaviour change policies and interventions could help improve their effectiveness and allow local authorities to build the evidence base of what works, when and for whom. This could help local authorities achieve policy objectives and, ultimately, improve the financial circumstances of their residents.

## Figures and Tables

**Figure 1 behavsci-13-00991-f001:**
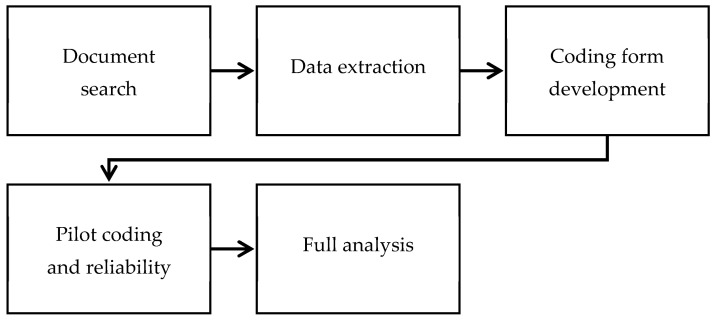
An overview of the key stages of the research.

**Table 1 behavsci-13-00991-t001:** A description of each behavioural area of interest.

Behavioural Area of Interest	Description
Claiming financial support	The take up of traditional financial support (e.g., state benefits) or other additional forms of financial support (e.g., free school meals, healthy start vouchers, local grants)
Paying bills	Paying essential bills (e.g., rent, utilities, council tax) on time to avoid arrears or other debt. This does not include debt management behaviours (i.e., paying off arrears)
Saving	Regular and appropriate saving to help protect against income shocks

**Table 2 behavsci-13-00991-t002:** The criteria that data fragments had to meet to be extracted for analysis under RQ1, RQ2 or RQ3/RQ4.

	Target Behaviours (RQ1)	Recognised Barriers and Enablers (RQ2)	Interventions (RQ3, RQ4)
Extraction criteria	The data must be attempting to describe a behaviour that would be performed by a council resident. The following definition was used as a guide: “Behaviours are physical events that occur in the body and are controlled in the brain” [35] The behaviour being described must be directly or indirectly linked to one of the three behavioural areas of interest. The behaviour must be described in the context of a behaviour change strategy (i.e., it must be the target behaviour) *	The data must be attempting to describe a barrier or enabler to a behaviour. Put simply, something that would inhibit or facilitate the performance of the behaviour. For example, a resident not knowing how to apply for benefits would be a barrier to them applying. Alternatively, a resident having high digital literacy skills would be an enabler to them making an online benefits application. The barrier or enabler must be directly or indirectly linked to one of the three behavioural areas of interest.	The data must be attempting to describe an intervention aiming to influence the behaviour of a resident. The following definition of behaviour change interventions was used as a guide: “Behaviour change interventions can be defined as coordinated sets of activities designed to change specified behaviour patterns.” [14] The intervention must be directly or indirectly linked with one of the three behavioural areas of interest. The council must have explicitly committed to implementing the intervention (e.g., within the “Action Plan” section of the document). Or the council must have already implemented the intervention (e.g., within the “Action to date” section of the document.

Note. * This criterion was met either by the intervention being described alongside the target behaviours (e.g., “the council will offer a monthly prize draw to encourage residents to pay rent via direct debit”) or by the context of the strategy being explained in another section of the document.

**Table 3 behavsci-13-00991-t003:** The codebook developed to apply the AACTT to any descriptions of target behaviours.

AACTT Domain	Definition [20]	General Example [20]	Allocated Codes [31]
Action	A discrete observable behaviour	Prescribing antihypertensives, providing a referral to a specialist, washing hands, setting a policy	Yes—There is an action that is discrete and observable in the statement No—There is no discrete and observable action (s) present in the statement
Actor	The individual or group of individuals who perform (or should/could) the Action	Primary care physician, pharmacist, social worker, resident, administrator, middle manager, head of unit, policymaker	Yes—Explicitly names the person OR persons responsible for performing the action No—No one is explicitly named to perform the action
Context	The physical, emotional or social setting in which the Actor performs (or should/could) the Action	Examination room, doctor’s office, outside a patient room, in a boardroom, stressful vs. calm situation, when patients’ relatives are present or not	Yes—A location/context for the action has been explicitly named No—A location/context for the action has not been explicitly named or assumptions need to be made to interpret the context. MIN—A location/context has been named but there could be multiple. OR a location/context has been named but would more information would be needed to interpret. Or when nonspecific terms are used to refer to the context that could have multiple meanings and have not been pre-defined.
Target	The individual or group of individuals for/with/on behalf of whom the Actor performs the Action	Patient with diabetes and blood pressure above 140/80 mmHg, patient wanting to quit smoking	Yes—An individual or group that the action is with/for and on behalf of is named. No—No individual or group that the action is with/for and on behalf of is named. OR assumptions have to be made to interpret who the Target is. MIN—When a reference to a Target is made but more information would be needed to interpret which specific individuals are the Target. Or one of with/for/behalf of is missing and should not be. NN—the Action does not require a Target or the Target is also the Actor.
Time	The Time period and duration that the Actor performs the Action in the Context with/for the Target	At annual review, next time a patient visits, every week, over the next 6 months	Yes—Time period and duration if relevant are explicitly specified or the situation when the action should occur is stated. No—No Time period or duration is specified or assumptions were made about them. MIN—When a reference to Time is made but more information would be needed to interpret or assumptions would need to be made to interpret or duration and/or frequency are present but not both.

Note. MIN = More information needed, NN = Not needed.

**Table 4 behavsci-13-00991-t004:** The codebook developed to identify COM-B components from the recognised barriers or enablers within the documents.

	COM-B Component	Definition [14,15]	General Example [14,15]	Criteria [11]
Capability	Psychological Capability	Knowledge or psychological skills, strength or stamina to engage in the necessary mental processes	Understanding the impact of CO_2_ on the environment	Code Psychological Capability if the barrier/enabler is related to an individual’s psychological capacity to engage in the target behaviour.
	Physical Capability	Physical skills, strength or stamina	Having the skills to take a blood sample	Code Physical Capability if the barrier/enabler is related to an individual’s physical capacity to engage in the target behaviour.
Opportunity	Physical Opportunity	Opportunity afforded by the environment involving time, resources, locations, cues, physical ‘affordance’	Being able to go running because one owns appropriate shoes	Code Physical Opportunity if the barrier/enabler is related to physical factors outside the individual that reduce the opportunity to engage in the target behaviour or prompt it.
	Social Opportunity	Opportunity afforded by interpersonal influences, social cues and cultural norms that influence the way that we think about things, e.g., the words and concepts that make up our language	Being able to smoke in the house of someone who smokes but not in the middle of a boardroom meeting	Code Social Opportunity if the barrier/enabler is related to social factors outside the individual that reduce the opportunity to engage in the target behaviour or prompt it.
Motivation	Reflective Motivation	Reflective processes involving plans (self-conscious intentions) and evaluations (beliefs about what is good and bad)	Intending to stop smoking	Code Reflective Motivation if the barrier/enabler is related to reflective brain processes that inhibit the target behaviour.
	Automatic Motivation	Automatic processes involving emotional reactions, desires (wants and needs), impulses, inhibitions, drive states and reflex responses.	Feeling anticipated pleasure at the prospect of eating a piece of chocolate cake	Code Automatic Motivation if the barrier/enabler is related to automatic brain processes that inhibit the target behaviour.
	Too vague to be coded			Code “Too vague to be coded” if the barrier/enabler cannot be coded reliably to any of the COM-B components

**Table 5 behavsci-13-00991-t005:** The codebook developed to identify ITs.

ITs	Definition [14,15]	General Example [14,15]	Relevant Example
Education	Increasing knowledge or understanding	Providing information to promote healthy eating	The council informing a resident of their entitlement to state benefits
Persuasion	Using communication to induce positive or negative feeling or stimulate action	Using imagery to motivate increases in physical activity	The council writing to residents with a campaign that uses emotive communication to promote saving
Incentivisation	Creating an expectation of reward	Using prize draws to induce attempts to stop smoking	The council offering a free monthly prize draw to tenants that pay rent via direct debit
Coercion	Creating an expectation of punishment or cost	Raise the financial cost to reduce excessive alcohol consumption	The council applying a late payment charge to tenants that do not pay their rent on time
Training	Imparting skills	Advanced driver training to increase safe driving	The council providing financial management training to residents.
Restriction	Using rules to reduce the opportunity to engage in the target behaviour (or to increase the target behaviour by reducing opportunity to engage in competing behaviours)	Prohibiting sales of solvents to people under 18 to reduce use for intoxication	The council implementing a policy that all rent must be paid by direct debit
Environmental restructuring	Changing the physical or social context	Providing on-screen prompts for GPs to ask about smoking behaviour	The council providing additional methods for tenants to pay their rent (e.g., direct debit, online, telephone)
Modelling	Providing an example for people to aspire to or imitate	Using TV drama scenes involving safe-sex practices to increase condom use	The council delivering a media campaign that shows a resident going online to search for the financial support available to them
Enablement	Increasing means/reducing barriers to increase capability (beyond training and education) or opportunity (beyond environmental restructuring)	Behavioural support for smoking cessation, medication for cognitive deficit, surgery to reduce obesity, protheses to promote physical activity	The council putting together a personalised budgeting plan for the resident
Too vague to be coded	Code “Too vague to be coded” if the recommendation is not explicit enough to be coded reliably.		The council stating they will work with residents to help them apply for benefits

**Table 6 behavsci-13-00991-t006:** The matrix that links the components of the COM-B model of behaviour to the ITs. Grey box indicates link.

	ITs								
**COM-B component**	Education	Persuasion	Incentivisation	Coercion	Training	Restriction	Environmental restructuring	Modelling	Enablement
Physical Capability									
Psychological Capability									
Physical Opportunity									
Social Opportunity									
Automatic Motivation									
Reflective Motivation									

Note. Adapted from previous research [14,15].

**Table 7 behavsci-13-00991-t007:** The number of data fragments that related to each behavioural area of interest.

Behavioural Area of Interest	No. of Data Fragments (%)
Claiming financial support	81 (34%)
Paying bills	146 (61%)
Saving	11 (5%)
Total	238

**Table 8 behavsci-13-00991-t008:** The number and percentage of the descriptions of target behaviour that met the Action, Actor, Context, Target and Time (AACTT) coding criteria.

				No. Descriptions of Target Behaviours							131		
Action		Actor		Context			Target				Time		
Meets criteria No. (%)	Does not meet criteria No. (%)	Meets criteria No. (%)	Does not meet criteria No. (%)	Meets criteria No. (%)	Partially meets criteria No. (%)	Does not meet criteria No. (%)	Meets criteria No. (%)	Partially meets criteria No. (%)	Does not meet criteria No. (%)	Not needed No. (%)	Meets criteria No. (%)	Partially meets criteria No. (%)	Does not meet criteria No. (%)
0 (0%)	131 (100%)	99 (76%)	32 (24%)	4 (3%)	11 (8%)	116 (89%)	0 (0%)	0 (0%)	0 (0%)	131 (100%)	4 (3%)	16 (12%)	111 (85%)

**Table 9 behavsci-13-00991-t009:** Examples of possible amendments to target behaviour descriptions to aid better specification and ensure they met the AACTT criteria.

Description of the Target Behaviour	Were the Criteria Met?					Possible Amendments to Meet All Criteria (Amendment in Bold)
	Action	Actor	Context	Target	Time	
“Deliver and promote digital skills training to residents, to teach residents how to: Search for discounts, Search for financial information and advice”—Blackpool, Financial Inclusion Strategy	No—‘search’ does not give enough detail on how the resident should be searching or what they should use to search.	Yes—the resident	No—where should the resident search for this information?	NN—not needed as the resident is acting on behalf of themselves	No—when should the resident search for this information?	Deliver and promote digital skills training to residents, to teach residents how to: Search for discounts and financial information/advice **using an online search engine at the library’s public computers when they come for an appointment with the council.**
“Before the signup stage, the Housing Officer will meet the prospective tenant and advise on the rent payment methods, and if applicable, to apply for Universal Credit or Housing Benefit.”—Dartford, Rent Arrears Policy	No—‘apply’ would involve several steps and there is no indication as to how the tenant should apply.	Yes—prospective tenant	No—where should the prospective tenant apply?	NN—not needed as the tenant would likely be applying on behalf of themselves	No—when should the prospective tenant apply?	Before the signup stage, the Housing Officer will meet the prospective tenant and advise on the rent payment methods, and if applicable, **to make an online application** for Universal Credit or Housing Benefit. **This should be done online, via telephone or at the Job Centre Plus offices within one week of signing the tenancy agreement.**
“Work with Credit Unions to promote and increase the uptake of their services”—Blackpool, Financial Inclusion Strategy	No—there is no action specified. What action needs to occur to increase the uptake?	No—who should be taking up the services of Credit Unions?	No—where should they be taking up the services of Credit Unions?	NN—not needed as the Actor would likely be acting on behalf of themselves	No—when should they take up the service of Credit Unions?	Work with Credit Unions to promote and increase **the number of residents in Blackpool that apply to open a savings account either online or in person when they start a new job.**
“Tenants who pay their rent weekly, need to pay on the Monday each week and tenants who pay it on a fortnightly or monthly basis need to pay in advance.”—East Devon, Income Management Strategy	No—how should be paying their rent each week?	Yes—tenants	No—where should they pay their rent?	NN—not needed as the tenant will likely be paying rent on behalf of themselves.	Yes—on the Monday each week or in advance.	Tenants who pay their rent weekly, need to pay on the Monday each week and tenants who pay it on a fortnightly or monthly basis need to pay in advance. **This should be paid via direct debit, via telephone or in person.**
“Work with GP surgeries and walk-in-centre services to support residents in completing application forms and to promote signposting to current service providers”—Blackpool, Financial Inclusion Strategy	No—how will the residents go about completing application forms? What application forms?	Yes—residents	MIN—It could be implied that the behaviour should take place in the GP surgery/walk-in-centre but this should be made explicit	NN—not needed as the resident is likely to be completing the Action on behalf of themselves.	No—when should the resident complete the application form?	Work with GP surgeries and walk-in-centre services to support residents in completing **online** application forms for **Universal Credit at the surgery whenever they come in for a check-up** and to promote signposting to current service providers

**Table 10 behavsci-13-00991-t010:** The number and percentage of descriptions of barriers/enablers that could be coded to each COM-B component.

COM-B Component	No. of Descriptions of Barriers/Enablers (%)	Example
Psychological Capability	8 (31%)	“Having poor financial knowledge or understanding can lead to costly credit or services which can reduce available income, leading to inability to budget and pay bills.”—West Lancashire, Financial Inclusion Strategy
Physical Capability	1 (4%)	“If the Council is aware that a tenant has difficulty in reading or understanding information given regarding their rent account and arrears, reasonable steps will be taken to ensure that the tenant understands any information given.”—Dartford, Rent Arrears Policy
Social Opportunity	0 (0%)	N/A
Physical Opportunity	19 (73%)	“High costs of living make it difficult for people on low incomes to meet basic costs”—Cambridge, Anti-Poverty Strategy
Automatic Motivation	0 (0%)	N/A
Reflective Motivation	0 (0%)	N/A
Too vague to be coded	4 (15%)	“Often, rent arrears are a symptom of underlying problems such as debt, illness or redundancy and if these problems can be addressed, then there is a good chance that rent arrears can be dealt with.” Dartford, Rent Arrears Policy

Note. The percentages will not add to 100% due to descriptions of barriers/enablers often being coded to more than one COM-B component.

**Table 11 behavsci-13-00991-t011:** The number and percentage of interventions where each IT was identified.

ITs	No. Interventions Where IT Was Present (%)	Example
Education	86 (46%)	“inform people of their entitlement to benefits, discounts, reliefs and exemptions where appropriate”—Bournemouth Christchurch and Poole, Debt Management Policy
Persuasion	1 (1%)	“Put in place customer facing documentation articulating RBC’s ‘support offer’. Explain the positivity’s of taking up support if offered and the consequences if refused.” (also coded to education and coercion)—Runnymede, Financial Wellbeing Strategy
Incentivisation	3 (2%)	“To encourage payment by direct debit there is a free monthly prize draw to those who pay by this method.”—Dartford, Rent Arrears Policy
Coercion	3 (2%)	“During this time the Housing Officer will: Remind the tenant of their responsibility to pay rent on time and explain the consequences of non-payment.” (also coded to education)—Dartford, Rent Arrears Policy
Training	11 (6%)	“Deliver and promote digital skills training to residents, to teach residents how to: Manage their money”—Blackpool, Financial Inclusion Strategy
Restriction	0 (0%)	N/A
Environmental restructuring	37 (20%)	“Promote and offer a wide variety of payment options”—Dartford, Rent Arrears Policy
Modelling	0 (0%)	N/A
Enablement	33 (18%)	“Our welfare benefit outreach workers: assist with applications”—Blackpool, Financial Inclusion Strategy
Too vague to be coded	52 (28%)	“Supporting low-income residents to access financial support”—Cambridge, Anti-Poverty Strategy

Note. The percentages will not add to 100% due to intervention descriptions often being coded to more than one IT.

**Table 12 behavsci-13-00991-t012:** The number and percentage of interventions where each BCT was identified.

Behaviour Change Technique (BCT)	No. Interventions Where BCT Was Present (%)	Example (Relevant Section Bolded)
Action planning	2 (1%)	“During this time the Housing Officer will: Remind the tenant of their responsibility to pay rent on time and explain the consequences of non-payment. **Check with the tenant how and when they will pay their rent and check which payment method will be used**” (also coded to ‘Social Support (practical)’)—Dartford, Rent Arrears Policy
Feedback on outcome(s) of behaviour	9 (5%)	**“Information will be provided about rent accounts through the issue of quarterly rent statements.”**—East Devon, Income Management Strategy
Social support (unspecified)	2 (1%)	“The Council **negotiate for competitive energy prices with the energy companies** on behalf of residents and make the switching process as simple as possible.”—East Riding of Yorkshire, Financial Inclusion Strategy
Social support (practical)	35 (19%)	“Where support issues are identified at a new tenancy sign-up, **we will refer tenants for local support** to either specialist council officers or Citizen Advice.”—Welwyn Hatfield, Rent Arrears Policy
Instruction on how to perform the behaviour	8 (4%)	“Deliver and promote digital skills training to residents, to **teach residents how to: Search for discounts, Search for financial information and advice**”—Blackpool, Financial Inclusion Strategy
Information about social and environmental consequences	5 (3%)	“Put in place customer facing documentation articulating RBC’s ‘support offer’. **Explain the positivity’s of taking up support if offered and the consequences if refused**.”—Runnymede, Financial Wellbeing Strategy
Prompts/cues	6 (3%)	“We may **send you a text message if you fall into arrears to prompt you to arrange payment**.”—Hinckley and Bosworth, Rent Arrears and Recovery Policy
Material incentive (behaviour)	2 (1%)	“The Council aims to maximise tenant’s income by **informing them of benefits they are entitled to**.” (also coded to ‘Information about social and environmental consequences’ and ‘Incentive (outcome)’—Dartford, Rent Arrears Policy
Material reward (behaviour)	2 (1%)	“Eligible tenants that have rent arrears due to the under-occupation size criteria **may be offered a financial incentive** to downsize to a smaller property”—Medway, Rent Arrears Recovery Policy
Incentive (outcome)	2 (1%)	“inform people of **their entitlement to benefits, discounts, reliefs and exemptions** where appropriate” (also coded to ‘Information about social and environmental consequences’ and ‘Material incentive (behaviour’)—Bournemouth Christchurch and Poole, Debt Management Policy
Future punishment	1 (1%)	“We also **explain the consequences of not paying rent**” (Welwyn Hatfield, Rent Arrears Policy)
Restructuring the physical environment	6 (3%)	“Lead a free school meal take-up campaign: (1) Extend the pilot programme **to simplify the** **application process** for parents/guardians.”—Plymouth, Child Poverty Action Plan
Adding objects to the environment	33 (18%)	“Promote and **offer a wide variety of payment options**”—Dartford, Rent Arrears Policy
Too vague to be coded	104 (56%)	“Supported benefit claimants to receive their full entitlement of Housing Benefit and Council Tax reduction.”—Cambridge, Anti-Poverty Strategy

Note. The percentages will not add to 100% due to intervention descriptions often being coded to more than one BCT.

**Table 13 behavsci-13-00991-t013:** The number and percentage of interventions that could be influencing each COM-B component.

COM-B Component	No. Interventions That Could Influence Component (%)
Psychological Capability	112 (61%)
Physical Capability	43 (23%)
Social Opportunity	63 (34%)
Physical Opportunity	71 (38%)
Automatic Motivation	69 (37%)
Reflective Motivation	87 (47%)

Note. The percentages will not add to 100% due to the matrix linking ITs to more than one COM-B component.

## Data Availability

The data presented in this study are listed within Appendix A.

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
