# Peer review of "Applying the Behaviour Change Wheel to UK Local Authority Policy Documents: A Content Analysis in the Context of Financial Behaviour"

_behavsci, 2023, doi:10.3390/bs13120991_

Round 1

Reviewer 1 Report

Comments and Suggestions for Authors

Dear authors

While the current introduction is acceptable, it would benefit from a stronger emphasis on addressing the gap and objectives outlined in the abstract: “ The extent to which these behaviour change activities are based on relevant theory or evidence is unknown. This research aims to retrospectively analyse the content of local authorities’ policies to identify opportunities for improvement.” Introduction appears to give an excessive amount of emphasis to methods.

A literature review section is missing.

The methodology should explicitly outline the reasons for selecting content analysis and clarify whether the approach is deductive or inductive. I believe that this reference could provide valuable insights:

Elo, S., & Kyngäs, H. (2008). The qualitative content analysis process. Journal of Advanced Nursing, 62(1), 107–115. https://doi.org/10.1111/j.1365-2648.2007.04569.x

While the study primarily focuses on the practical implications, which is ok, it is also essential to also provide theoretical implications of their research.

Good luck with you work.

Reviewer 2 Report

Comments and Suggestions for Authors

the topic is interesting, following points need to be improved,

the introduction could be shortened and focus on the innovation or marginal contribution of the work. 

the literature review could be separated from the introduction part, and it is better to illustrate the research question one by one with fully explanation. 

the organization of the manuscript is not well, please reorganize the structure in author-friendly way. 

Comments on the Quality of English Language

generally, the quality of English is ok.

Reviewer 3 Report

Comments and Suggestions for Authors

The article is generally well-structured and describes exciting research in content and methodology. The title and the abstract are well-phrased. Using the COM-B model to identify any barriers or enablers to targeted behaviors within policy documents is a novelty of the research. The examples throughout the text make it easier to follow and elevate the value of the paper.

Methods section: It is unnecessary to refer to Microsoft Excel explicitly. The point is that a database was used. Please rephrase.

Furthermore, a step-by-step overview of the methods used would be beneficial. It could be a figure or a longer paragraph.

Table 2 column RQ2 should be supplemented with a description of enablers.

Table numbering and page numbering need to be clarified. Reference in the text to tables needs to be reviewed.

The table titled: The codebook developed to identify COM-B components… could be visually separated into three parts (capability, opportunity, and motivation).

It would be nice to know how the coders were selected for the research.

Implications: Data fragments related to saving are much lower than for claiming financial support and paying bills (Table 6). Please include some policy implications on this finding.

Recommendation for future research: Behavioral areas from CFPB and MaPS should be included in extended research.
